# Characterization of the Gut Microbiota in Individuals with Overweight or Obesity during a Real-World Weight Loss Dietary Program: A Focus on the Bacteroides 2 Enterotype

**DOI:** 10.3390/biomedicines10010016

**Published:** 2021-12-22

**Authors:** Rohia Alili, Eugeni Belda, Odile Fabre, Véronique Pelloux, Nils Giordano, Rémy Legrand, Pierre Bel Lassen, Timothy D. Swartz, Jean-Daniel Zucker, Karine Clément

**Affiliations:** 1Nutrition and Obesities, Systemic Approaches, NutriOmics Research Unit, INSERM, Sorbonne Université, 75013 Paris, France; rohia.alili@aphp.fr (R.A.); veronique.pelloux-gervais@upmc.fr (V.P.); pierre.bellassen@aphp.fr (P.B.L.); jean-daniel.zucker@ird.fr (J.-D.Z.); 2Integrative Phenomics, 75011 Paris, France; nils.giordano@integrative-phenomics.com (N.G.); timothy.swartz@integrative-phenomics.com (T.D.S.); 3Groupe Éthique et Santé, 13400 Aubagne, France; odile.fabre@groupethiquetsante.com (O.F.); remy.legrand@groupeethiquesante.com (R.L.); 4UMMISCO, Unité de Modélisation Mathématique et Informatique des Systèmes Complexes, IRD, Sorbonne Université, F-93143 Bondy, France; 5Nutrition Department, Pitié-Salpêtrière Hospital, Assistance Publique-Hôpitaux de Paris, 75013 Paris, France

**Keywords:** obesity, gut microbiota, weight loss, Bacteroides 2 enterotype, real-world dietary intervention, nanopore technology

## Abstract

Background: Dietary intervention is a cornerstone of weight loss therapies. In obesity, a dysbiotic gut microbiota (GM) is characterized by high levels of Bacteroides lineages and low diversity. We examined the GM composition changes, including the Bacteroides 2 enterotype (Bact2), in a real-world weight loss study in subjects following a high-protein hypocaloric diet with or without a live microorganisms (LMP) supplement. Method: 263 volunteers were part of this real-world weight loss program. The first phase was a high-protein low-carbohydrate calorie restriction diet with or without LMP supplements. Fecal samples were obtained at baseline and after 10% weight loss for 163 subjects. Metagenomic profiling was obtained by shotgun sequencing. Results: At baseline, the Bact2 enterotype was more prevalent in subjects with aggravated obesity and metabolic alterations. After weight loss, diversity increased and Bact2 prevalence decreased in subjects with lower GM diversity at baseline, notably in LMP consumers. Significant increases in *Akkermansia muciniphila* and *Parabacteroides distasonis* and significant decreases of *Eubacterium rectale*, *Streptococcus thermophilus* and Bifidobacterial lineages were observed after weight loss. Conclusions: Baseline microbiome composition is associated with differential changes in GM diversity and Bact2 enterotype prevalence after weight loss. Examining these signatures could drive future personalized nutrition efforts towards more favorable microbiome compositions.

## 1. Introduction

Overweight and obesity are global health concerns with the number of obesity cases having tripled since 1975, resulting in 39% and 13% of adults being overweight or obese, respectively, in 2016 [1]. An elevated Body Mass Index (BMI) is also a major risk factor for type 2 diabetes, dyslipidemia, hypertension, cardiovascular and liver diseases, kidney failure, osteoarthritis, and several cancers [2,3]. Importantly, a 5% reduction in body weight can significantly improve metabolic functions and decrease comorbidity risks [4]. Lifestyle and diet changes are cornerstones of weight loss management [3]. However, individual weight loss responses to diet and lifestyle changes are highly variable, and the effectiveness of dietary interventions could be improved by identifying the underlying factors behind this variability.

The gut microbiota (GM), known for its role in health and metabolic diseases, is strongly influenced by dietary intake [5] and is increasingly suggested as an important factor contributing to individual dietary responses [6,7,8]. Obesity is characterized by GM alterations resulting in dysbiosis. This includes a loss of microbial diversity, and changes in composition, both of which are correlated with host metabolic and inflammatory perturbations. Moreover, increasing BMI into severe obesity is associated with further microbiota diversity loss [6,9]. At taxonomic levels, an altered ratio between the Firmicutes and Bacteroidetes phyla has been proposed as a signature of obesity, although with conflicting reports. For example, both low and high Firmicutes/Bacteroides ratios have been reported in obese subjects [10] (and references therein). Substantial temporal variability in human gut microbiome profiles could partially explain those conflicting reports in the association of metagenomic features with obesity, which could be resolved with repeated measurement designs as used for robust biomarker discovery. Analyses can also be focused on community-wide microbiome descriptors and indices [11].

Some of these descriptors are the GM enterotypes, which are discrete classifications of individual microbiome compositions [12]. For instance, the Bacteroides 2 (Bact2) enterotype is a specific microbiome composition enriched in the relative abundance of Bacteroides genera. Bact2 has been associated with an overall low microbial cell density and successfully associated with obesity severity [13,14]. Given the high complexity of GM community compositions, both identifying and using enterotypes as tools to describe the functional activity of the microbiome have been controversial in the field of metagenomics [15]. Nonetheless, the Bact2 dysbiotic enterotype may stand out because, in addition to its association with obesity [9], it is more prevalent in several inflammatory-related diseases including ulcerative colitis [13], inflammatory bowel disease and primary sclerosing cholangitis [16]. The Bact2 enterotype is thus an emerging but robust signature of a dysbiotic microbiome, yet its relevance during weight loss and dietary intervention has not been examined.

Lifestyle [6], pharmaceutical, and even surgical weight-loss interventions have a strong and rapid influence on GM composition and functionality [17]. Specifically, dietary intervention-induced weight loss increases microbiome diversity, richness, and the abundance of beneficial microbial species, such as *Akkermansia municiphila*, *Lactobacillus*, or *Bifidobacterium* [6,18,19]. Moreover, our group and others reported that pre-interventional GM features can also influence GM responses and host metabolic outcomes during a dietary intervention [6,9,19]. In addition, the use of microbiome-targeted supplements, an umbrella term including probiotics, prebiotics, or synbiotics, can also modulate GM composition and eventually improve metabolic health [20,21].

Despite the pre-clinical and clinical evidence demonstrating the link between diet, microbiome, and weight loss, there is a general lack of large-scale longitudinal real-world evidence. Thus, the goal of this study was to answer five major questions: (1) What are the GM compositional characteristics in a real-world population with overweight and obesity, and how do these characteristics relate to metabolic health parameters before weight loss? (2) Is there a relationship between the dysbiotic Bact2 microbiome composition and subjects’ clinical features? (3) Are these GM features affected by weight loss induced by a hypocaloric high-protein, low-glycemic index diet? (4) Are GM baseline characteristics associated with GM changes post-weight loss and/or with weight loss-related clinical outcomes? (5) Does the use of a live microbial product (LMP) improve weight-loss outcomes and/or provide benefits to GM composition during a high-protein calorie-restricted diet?

Here, we address these questions by leveraging a real-world French study on volunteers with overweight or obesity following a standardized medical and dietary weight-loss program as part of standard care.

## 2. Subjects and Methods

### 2.1. Overall Dietary Program

#### 2.1.1. Nutritional and Psycho-Behavioral Reeducation (RNPC) Program

The RNPC program (Rééducation Nutritionnelle et Psycho-Comportementale) is a weight loss program developed by a healthcare provider and organized as a network across 13 regions in France. This is a real-world weight loss intervention that was developed based on outcomes of a clinical intervention from the European DIOGENES consortium [22]. The RNPC program is conducted by a registered dietitian in each center. The majority of patients with overweight and obesity participating in the RNPC program are referred to RNPC centers by their primary care physicians (PCP) for weight management, but patients can also join the program without being directly referred by their physician.

The care provider (RNPC) proposes a weight loss program in three phases that is standardized across all centers. Phase 1 consists of 8 weeks during which individuals follow a high-protein, low-carbohydrate calorie-restricted diet, including the intake of 3 to 5 commercial food products enriched in protein; the number of proposed food supplement depend on an individual’s lean body mass percentage and initial body weight. During this first phase, calorie intake is standardized to 800 kcal for women and 1000 kcal for men. At the end of phase 1, the subjects start a stabilization phase (phase 2) during which the protein supplements are gradually reduced and a normal diet is reintroduced while continuing to emphasize high protein and low glycemic index carbohydrate intakes. Phase 3 is the weight maintenance phase. This regimen has been previously described in detail [23].

Based on these three standardized phases, the registered dietitian recommends varying quantities of vegetables, animal proteins (meat, fish, eggs, or shellfish), and food supplements rich in protein (in the form of cookies, cereal bars, bread, crackers, soups, omelets, drinks, and desserts). During phase 1 (weight loss), these dietary supplements contribute to 40% of the total energy intake. Each 30 g serving provides approximately 110 kcal and contains 15.8 g of protein, 2.4 g of carbohydrates, 5.4 g of fat, and 2.3 g of fiber. Supplements are enriched with vitamins and minerals to avoid possible deficiencies due to the caloric restriction. These products comply with the requirements specified by the European Food Safety Authority (EFSA) Panel on Dietetic Products Nutrition and Allergies (NDA), 2015 [24]. During phase 2 (weight stabilization phases), the intake of food supplements gradually decreases and is replaced by whole food. At the end of the stabilization period, the target macronutrient composition is 25% protein, 45% carbohydrate, and 30% fat. Throughout all phases, participants are recommended a live microbial product (LMP), which is a food-grade supplement containing a mixture of 11 living bacteria strains and one yeast species with the following composition *Lactobacillus rhamnosus* ATCC n. 53103 (lactose): 2.5 × 10^9^ UFC, *Lactobacillus reuteri* DSM 23878 (lactose): 2.5 × 10^9^ UFC, *Lactobacillus gasseri* BIO6369: 6.25 × 10^9^, *Lactobacillus acidophilus* W22: 3.9 × 10^5^ UFC, *Lactobacillus plantarum* W21: 3.65 × 10^5^ UFC, *Lactococcus lactis* W19: 3.65 × 10^5^ UFC, *Lactobacillus paracasei* W20: 2.8 × 10^5^ UFC, *Lactobacillus salivarius* W24: 2.8 × 10^5^ UFC, *Bifidobacterium lactis* W51: 2.8 × 10^5^ UFC, *Bifidobacterium lactis* W52: 0.56 × 10^6^ UFC, *Enterococcus faecium* W54: 2.8 × 10^5^ UFC and 100 mg de *Saccharomyces cervisiae,* This food supplement also contains inulin (27.6 mg), and fructooligosaccharides (2.40 mg). The RNPC LMP supplement is commercialized under the name “MGF” (for *Megaflora*), which is a combination of different products and ingredients: Megaflora9, Rhamnosus LRGG, *Lactobacilus reuteri* LRE02, BoulardII, *Lactobacililus gasseri* LGS06.

Subjects are followed in each center by a registered dietitian every two weeks who measures weight, waist circumference, and body fat mass by impedancemetry (Beurer BG42, Ulm, Germany) during each visit, a blood panel is prescribed by their PCP to evaluate glycemia, insulinemia, triglycerides, and liver enzyme before the intervention and after at least 10% weight loss.

#### 2.1.2. The GutInside Study

We took advantage of the standardized real-world intervention in RNPC centers to examine the GM profile of volunteers before and after 10% weight loss. In a registered protocol (NCT04822948), dietitians (as investigators) recruited patients before starting the RNPC program. All subjects signed informed consent before participating in the study. During their first visit, volunteers were requested to collect their stool sample before starting phase 1 and when 10% weight loss was reached, generally occurring during the first weight loss phase 1 (i.e., a hypocaloric high protein and low glycemic index carbohydrate diet). Otherwise, the subjects’ follow-up was the same as those not participating in the GutInside study. Subjects were recruited from September 2018 to January 2020 in 64 French RNPC centers. The anthropometric characteristics of each participant were measured to determine body composition (by Beurer BG42 impedancemetry device) at baseline and at a 10% weight-loss period. A 12 h fasting blood sample was drawn and parameters relating to glucose and lipid homeostasis, and hepatic and kidney function were evaluated at the two time points. Inclusion criteria were as follows for subjects: age between 18 and 65 years old, a BMI ≥ 25 kg/m^2^, and having signed informed consent in an RNPC center. The exclusion criteria were pregnancy, breastfeeding, antiretroviral therapy, renal insufficiency, severe hepatic disease, and anemia (<10 g/dl). Subjects with known gastrointestinal illness, having undergone bariatric surgery, presenting a weight loss of more than 10% of body weight during the last 3 months, and taking antibiotics, prebiotics, probiotics, or synbiotics, were also excluded.

### 2.2. Sample Collection and Bacterial DNA Extraction

A total of 426 stool samples were collected at home by 263 volunteers before starting the RNPC weight loss program (V1). For 163 of the 263 subjects, fresh stools were also collected after 10% of weight loss (V2). Stool samples were collected in Zymo DNA/RNA Shield-Fecal Collection Tube (Ozyme), which allows up to 2 g of stool sample to be collected. The tubes were sent by the French postal system and stored at room temperature at the sequencing center, according to manufacturers’ directions. Bristol stool score was used to assess stool consistency.

Bacterial DNA extraction was performed using, “Pure Link ™ Microbiome DNA Purification Kit” (Invitrogen, Paris, France) with the optimized protocol developed in our lab [25]. The DNA yield was evaluated by a fluorometer, Qubit (Life Technologies Alfortville, France), and DNA quality was evaluated by Nanodrop (Thermo Scientific, Alfortville, France) and Tape station (Agilent, Les Ulis, France).

### 2.3. Library Preparation and Sequencing

A total of 1.5 µg of DNA was used to perform PCR-free library construction. DNA end repair was performed using the NEBNext FFPE Repair Mix (New England Biolabs (NEB), Evry, France). We used NEBNext Ultra II End Repair/dA-Tailing Module (NEB) for the “end prep” step, 1D Native barcoding genomic DNA kit (Oxford Nanopore Technologies (ONT)), and “NEB Blunt/TA Ligase Master Mix kit (NEB) for DNA multiplexing and adapters ligation. Agentcourt AMPure XP (Beckman Coulter, Villepinte, France) beads were used for DNA purification.

Whole genome metagenomic sequencing was performed with a MinION sequencer (ONT) using 48 h runs and 12 samples per run. A rarefaction threshold of 10,000 reads per sample was used to perform metagenomic analysis.

### 2.4. Bioinformatics Treatment

Samples were sequenced over 35 runs with a total of 332.54 million reads generated (average 5.54 million reads per run) with an average size of 2.8 kilobases per read. Reads were base-called, quality filtered, and demultiplexed with Guppy (Version 2.1.3). In each run, at least 78.87% of sequencing reads could be confidently assigned to different sample barcodes (426,271.4 reads per sample). Taxonomic binning of Nanopore reads was carried out in a two-step procedure. In a first step, we used Centrifuge [26] to remove human contaminants and classify reads with an NCBI reference taxonomic identifier, using the precompiled “P_compressed_b + v + h” database (shipped with Centrifuge) as a reference database. In a second step, the classified sequences were mapped against the corresponding reference genome sequence using Minimap2 with the map-ont option that is optimized for Oxford Nanopore reads [22] in order to filter out false-positive taxonomic assignments. Species-level abundance tables were built from taxonomic binning results and were integrated with taxonomic information and sample metadata in *phyloseq* R objects [27] for subsequent ecological analyses in R.

Alpha diversities (Observed species, Shannon) were computed from the rarified species abundance table with the estimated richness function of the phyloseq R package [28]. The vegan R package was used to compute Beta-diversity matrices from the rarified species abundance table collapsed at genus level (vegdist function), and to visualize microbiome similarities with principal coordinate analysis (PCoA) (cmdscale function) [29]. Enterotype classifications were performed from the genus abundance matrix using the Dirichlet multinomial mixture (DMM) method as described in Holmes and al. [30] and implemented in the Dirichlet Multinomial R package. The estimation of the explanatory power of clinical features regarding relative, genus-level, microbiome profiles variation was performed using univariate or multivariate stepwise distance-based redundancy analysis as implemented in the vegan R package [31].

The abundance of KEGG ortholog groups (KO groups) was quantified from species-level pan-genome KO tables built from reference genomes in the Centrifuge database, from which the abundance of KO was computed as the sum of the abundances of the species containing these KO groups. Gut metabolic modules (GMMs) were computed from the KO abundance matrix using the GOmixer R package [32].

### 2.5. Statistical Analyzes

Statistical analysis was performed in R v3.6.2. Taxonomic and functional features with less than 5 × 10^−4^ relative abundances in less than 20% of the samples were excluded from the statistical analysis. Associations of clinical variables with species richness and Bact2 status were tested with linear and logistic regressions, respectively, while adjusting for different confounders (age, gender, center). Linear mixed-effect models LMM was used to test for differences in taxonomic and functional features between baseline and 10% weight loss with an individual as a random factor. This was carried out separately for LMP consumers and LMP non-consumers (Appendix A) to compare the effects of LMP supplementation on metagenomic features and by adding LMP intake as a second random factor in order to control for variability introduced by LMP intake in the temporal response of interest in each case (all follow-up cohort and follow-up cohort stratified by baseline diversity, respectively). All statistical tests used were two-tailed. All *p*-values were corrected for multiple testing as appropriate using the Benjamini–Hochberg procedure. Adjusted *p* values < 0.05 were reported as significant. The Kruskal–Wallis test with post hoc Dunn test was used to test for differences between taxonomic and functional metagenomic features across enterotypes. Chi-square test and Kruskal–Wallis test were used to test for differences between categorical and clinical variables respectively across enterotypes. Clinical differences between LMP consumer and non-consumer groups on baseline were evaluated with Student’s *t*-test for continuous data and chi-square tests or Fisher’s exact tests for categorical data. Clinical differences at 10% weight loss were evaluated with paired Student’s *t*-tests.

The propensity scores method [33] was used to define equally-sized groups of LMP consumers and non-consumers (*n* = 47 individuals per group) matched by age, gender, and weight with the Matching R package [34].

## 3. Results

### 3.1. Clinical and Biological Variables of the Studied Population

A total of 208 women and 55 men participated in the GutInside real-world weight loss intervention study. The average participant age was 50 years old with subjects in the following BMI categories: Four subjects with BMI = 25 kg/m^2^, 74 subjects with 25 < BMI ≤ 30 kg/m^2^, 112 subjects with 30 <BMI ≤ 35 kg/m^2^ and 73 with a BMI >35 kg/m^2^. The time needed to achieve the 10% weight loss target ranged from 27 to 180 days. Of the 263 individuals participating in the study, 188 consumed the LMP supplementation (consumers) while 75 followed the program without any supplementation (non-consumers). Baseline parameters of all participants are shown in Table 1. Except for fasting blood glucose and gamma glutamyl-transpeptidase (GGT), there were no statistically significant differences between the two groups for baseline clinical parameters (Table 1).

### 3.2. Participants’ Gut Microbial Composition, the Bact2 Enterotype, and Metabolic Profile at Baseline

We examined the relationship between clinical and biological variables and fecal microbiota composition at baseline (i.e., before the dietary program) using a univariate dbRDA with all variables (*n* = 45) and then a multivariate model building an ordiR2 function with a minimal number of variables. Across the 45 clinical and biological variables examined, 10 showed a significant link with microbiome composition in univariate dbRDA (*p*-value < 0.05; Figure 1a). In this effect size analysis, Bristol Stool Score (BSS) explained the largest part of the variance of genus-level microbiome composition (R^2^ = 1.84%, *p*-value = 0.003), followed by bodyweight (R^2^ = 1.16%, *p*-value = 0.001), alanine-aminotransferase levels (ALAT, R^2^ = 1.08%, *p*-value = 0.003) and obesity status (R^2^ = 1.03%, *p*-value = 0.003). Only BSS, ALAT and obesity status were found to explain a non-redundant fraction of microbiome variation based on stepwise dbRDA analysis (R^2^ = 1.51% (*p*-value = 0.006), 1.14% (*p*-value = 0.07), 0.59% (*p*-value = 0.06), respectively; Figure 1a).

We found a negative association between observed bacterial species (microbial diversity) and stool moisture as assessed by the Bristol stool score (Figure 1b, *p*-value = 8.62 × 10^−3^, beta = −0.49), which is in agreement with previous findings in other populations [13,14]. Negative associations with microbial diversity were also found for corpulence variables which included bodyweight (*p*-value = 3.78 × 10^−3^), BMI (*p*-value = 7.51 × 10^−3^; beta = −0.16) and severe obesity (*p*-value = 5.5 × 10^−3^; beta = −0.63 morbid obesity vs. non-obese status), as well as with uricemia (*p*-value = 0.033; beta = −0.16) and ALAT (*p*-value = 0.025; beta = −0.15) (Figure 1b). Controlling for age and recruiting centers did not change the nature of these associations (Figure 1b) but revealed additional associations between decreased species richness with fat mass (*p*-value = 0.018; beta = −0.17), obese status (*p*-value = 0.042; beta = −0.29 Obese vs. Non-obese status) or sleep apnea (*p*-value = 0.036; beta = −0.39). Similar associations were observed with evenness index (Shannon index; Figure 1b).

We next performed enterotyping on abundance profiles at the genus level using DMM models [30], which allowed us to group samples by microbial composition similarity. This approach uncovered an optimal stratification of four genus-level enterotypes (Appendix A) characterized by an abundance of *Faecalibacterium* genera of the *Ruminococcaceae* family (Rum), Bacteroides genus in groups Bacteroides 1 (Bact1) and Bacteroides 2 (Bact2) and Bifidobacterium genus (Bif) (Appendix A). This stratification of four groups reproduces previous studies [13,14,30]. However, in the present cohort, one of the enterotypes was enriched in Bifidobacterium instead of Prevotella lineages (Appendix A). Species richness and evenness differed across the four enterotypes. Importantly, the Bact2 enterotype, a composition characterized by the highest levels of Bacteroides genus abundances (Appendix A) showed the lowest levels of species richness and evenness (Appendix A) which confirms the dysbiotic characteristic of this microbiome composition, in agreement with previous studies [13,14]. In contrast, Rum composition, characterized by the highest levels of *Faecalibacterium* genera (Appendix A) shows the highest levels of species richness among the four microbiome compositions (Appendix A), but this is not translated to high levels of species evenness (Appendix A).

Univariate analyses over clinical and demographic covariates showed significant associations of enterotype composition with age and gender, with Bifidobacterium composition being younger (Appendix A; *p*-value < 0.05) and enriched in women (Appendix A; *p*-value < 0.05). Enterotype composition was also associated with corpulence variables such as obesity status, weight, BMI, and waist circumference (Appendix A), as well as ALAT and the prescription of Sleep apnea device (Appendix A), showing that Bact2 composition associates with a worst clinical profile in all cases.

This was confirmed in a logistic regression framework controlling for cofounding variables, where we found that the prevalence of low microbial diversity was the highest in the Bact2 enterotype (Figure 2a; binomial logistic regression adjusted by age, gender, and center of recruitment; *p*-value = 4.06 × 10^−12^; relative risk = 0.98, where relative risk can be interpreted as the scale factor necessary to obtain the prevalence of the Bact2 enterotype after a unit increase in species richness). Further analysis of the clinical variables showed that the Bact2 group had a more severe metabolic phenotype. The prevalence of the Bact2 enterotype increased with BMI (*p*-value = 2.7 × 10^−3^; relative risk = 1.08), weight (*p*-value = 1.51 × 10^−3^; relative risk = 1.03) and waist circumference (*p*-value = 0.02; relative risk = 1.02), as well as with insulin resistance (*p*-value = 0.011; relative risk = 1.19) and fasting insulin (*p*-value = 0.01; relative risk = 1.07). The Bact2 enterotype was also more prevalent in obese subjects with BMI ≥ 35 kg/m^2^ (*p*-value = 6.38 × 10^−3^; relative risk = 3.08), subjects with sleep apnea (*p*-value = 0.047; relative risk = 1.95), or subjects using a continuous positive airway pressure (CPAP) device (*n* = 25), compared to untreated subjects (*n* = 12) (*p*-value = 0.047; relative risk = 2.05) (Figure 2b). Despite methodological differences, these findings confirm previous observations made in the European MetaCardis cohort, where a relative risk of 1.05 was reported for the association between Bact2 status and BMI, confirming the links between Bact2 enterotype and aggravated obesity and comorbidities [14]. Finally, similar results were observed when the enterotyping procedure was carried out from species-level abundance data in terms of best stratification of the community around 4 discrete compositions (Appendix A), including a dysbiotic microbiome composition with low levels of species richness that correspond to Bact2 samples characterized by enterotyping at the genus level (Appendix A) and associated with a worse clinical profile (Appendix A).

### 3.3. Bacteroides 2 Enterotype, Taxonomy, and Functional Features and Obesity Severity

We examined quantitative taxonomic and functional characteristics of the metagenomic features associated with Bact2 microbiome composition. We found an enrichment of 8 bacterial species (Figure 3a; FDR < 0.05 in pairwise comparisons of Bact2 vs. all other enterotypes) including *Bacteroides fragilis* (FDR = 6.14 × 10^−16^) and *Bacteroides vulgatus* (FDR = 1.58 × 10^−13^). We observed that the Bact2 enterotype showed the lowest abundance of *F. prausnitzii* (FDR = 2.43 × 10^−25^). At the functional level, the Bact2 composition was enriched in 12 functional modules (Figure 3b; FDR < 0.05 in pairwise comparisons of Bact2 vs. all other enterotypes). Several of the functional modules were related to amino acid degradation, central metabolism, and degradation of carbohydrates. When put in relation with clinical variables, we observed positive significant associations between Bact2-enriched taxonomic and functional features and BMI that was robust to different covariate adjustments (Figure 3c,d; FDR < 0.05, standardized beta-coefficient > 0; linear regression models of feature abundance ~ BMI + additional covariates). Overall, bacterial signatures and functional characteristics linked to the Bact2 enterotype were also associated with obesity severity.

### 3.4. Gut Microbiota Diversity Evolution during Weight Loss Is Influenced by Baseline Gut Microbiome Diversity

For most subjects (84%), 10% weight loss was achieved during the calorie-restrictive weight loss phase 1, with a median duration of 75 days. Analyses were focused on the 163 subjects with paired stool sampling at the start of the calorie restriction and at 10% weight loss. Table 2 shows the changes in clinical and biological variables after 10% weight loss. As expected, the dietary intervention led to significant improvements in bodyweight, waist circumference, fat mass, and biological variables (fasting blood glucose, cholesterol, HbA1c, ALAT, GGT) compared to baseline. No significant differences were observed between the individuals based on LMP supplement intake (i.e., consumers and non-consumers groups) (Appendix A). The first set of analysis was then performed grouping both groups or examining LMP supplement consumers vs. non-consumers.

In the whole cohort, no significant evolution of microbiome diversity was observed after 10% weight loss (univariate test, *p*-value = 0.1). However, previous studies with lifestyle or surgical weight-loss interventions showed that such an evolution might be related to baseline microbiome diversity [6,9,19]. This motivated us to stratify our cohort based on the baseline diversity: we used the median bacterial species richness (490 observed species) as a threshold to group subjects into “high” (N = 74) and “low” (N = 89) diversity groups. When comparing gut microbiome diversity evolution between groups during the intervention, subjects from the low-diversity group exhibited significantly higher improvements of their microbiome diversity compared to subjects from the high-diversity group (Figure 4a; *p*-value = 1.35 × 10^−6^ on a comparison of species richness fold-change between baseline and 10% weight loss). Similar dynamics of diversity changes were observed with evenness indexes (Shannon index; Figure 4a). In addition, we observed that baseline low-diversity subjects had significantly higher levels of hepatic transaminases ALAT and ASAT (Appendix A), softer stools (Appendix A), and a trend towards higher weight than subjects in the high-diversity group (*p*-value = 0.073, Appendix A). We confirm that stratifying by baseline GM diversity led to different changes in diversity during a weight loss intervention. Given the link between the Bact2 enterotype and low microbiome diversity, this finding prompted us to address the effect of baseline diversity on the potential of an enterotype switch of the subjects at 10% weight loss.

### 3.5. Bacteroides 2 Prevalence after Weight-Loss

In the high-diversity group, we observed significant changes in enterotype composition at 10% weight loss (*p*-value = 0.0041). Noteworthy, 12 individuals of this group ended up switching to Bact2 at 10% weight loss (Figure 4b). However, no significant change in overall microbiome composition at 10% weight loss was observed (Figure 4c; R^2^ = 0.75%, *p*-value = 0.33, based on genus-level relative abundance profiles). An evaluation of the clinical changes in the 12 individuals that switched to Bact2 showed an increase in their blood uricemia, which contrasted with the other individuals of the high-diversity baseline group that predominantly exhibited a decrease in uricemia with weight loss (*p*-value = 0.033 Appendix A).

In the low-diversity group, we also observed significant changes in enterotype composition (Figure 4b, *p*-value = 0.0015) but with a different pattern. The overall microbiome composition was significantly altered after the intervention (Figure 4d; R^2^ = 1.2%, *p*-value = 0.041, Permanova based on genus-level relative abundance profiles). Weight loss led to significant changes in enterotype composition that mainly switched to Bact1 from Bif, Rum, and Bact2 compositions. As a result, Bact1 prevalence increased from 20% at baseline to 43% after 10% weight loss (Figure 4b). The prevalence of Bact2 decreased from 40% to 28% (Figure 4b) with 16/36 (44%) Bact2 individuals switching to Bact1 and 14/36 (39%) individuals remaining in the Bact2 group.

Overall, 25 of the 89 individuals in the low-diversity group were characterized by the Bact2 enterotype at 10% weight loss (Figure 4b). No change in uricemia was seen in this group. However, while they showed an improvement in glycemic status at 10% weight loss in comparison with non-Bact2 low-diversity individuals (Appendix A; *p*-value = 0.0024), (Appendix A), they remained with significantly higher levels of HbA1C compared with non-Bact2 individuals (Appendix A, *p*-value = 0.022) after weight loss. This again illustrates how the Bact2 microbial composition is associated with a deleterious metabolic status.

In summary, low microbiome-diversity individuals with baseline enrichment in Bact2 microbiome composition showed better improvement of their microbiome diversity, mainly by switching towards the Bact1 enterotype after a calorie restriction intervention. Changes were limited in the high-diversity group.

### 3.6. Bacterial Species after Weight Loss; Different Pattern with Diversity Stratification

Next, we examined the changes in individual bacterial species after weight loss. The abundance of 13 microbial species showed significant changes in abundance after 10% weight loss (Appendix A, FDR < 0.05, linear mixed effects models of feature abundance by intervention time, with patient and LMP supplementation intake as random factors). Among species that increased in abundance (beta-coefficients > 0 in Appendix A), *Akkermansia muciniphila* increased the most (beta-coefficient = 0.03, FDR = 5.62 × 10^−4^), followed by *Parabacteroides distasonis* (beta-coefficient = 0.025, FDR = 7.23 × 10^−3^), *Methanobrevibacter smithii* (beta-coefficient = 0.013, FDR = 0.02), and *Intestinimonas butyriproducens* (beta-coefficient = 8.99 × 10^−3^, FDR = 0.026). Among species decreasing in abundance, *Eubacterium rectale* decreased the most (beta-coefficient = −0.042, FDR = 1.42 × 10^−4^), followed by actinobacterial lineages such as *Bifidobacterium adolescentis* (beta-coefficient = −0.023, FDR = 0.036) or *Bifidobacterium bifidum* (beta-coefficient = −0.014, FDR = 0.02), and firmicutes lineages such as *Streptococcus thermophilus* (beta-coefficient = −0.023, FDR = 1.42 × 10^−4^) or *Streptococcus salivarius* (beta-coefficient = −0.012, FDR = 8.93 × 10^−3^).

On the functional level, the abundance of 16 gut metagenomic modules (GMMs) significantly changed after 10% weight loss (Appendix A; FDR < 0.05, linear mixed effects models of feature abundance by intervention time, with patient and LMP supplementation intake as random factors). GMMs increasing in abundance were acetate to acetyl-CoA conversion (beta-coefficient = 0.0036, FDR = 0.038), rhamnose and mannose degradation (beta-coefficient = 2.9 × 10^−3^, 2.8 × 10^−3^, FDR = 0.031, 0.018 respectively), and modules involved in methanogenesis that can be linked to methanogenic archaeon *M. smithii* (Appendix A). GMMs decreasing in abundance were linked to cysteine biosynthesis (beta-coefficient = −4.15 × 10^−3^, FDR = 0.25), sulfate reduction (beta-coefficient = −4.05 × 10^−3^, FDR = 0.039), degradation of fibers such as fructan and starch (beta-coefficient = −3.62 × 10^−3^, −3.08 × 10^−3^, FDR = 0.019, 7.86 × 10^−2^, respectively), and butyrate production (beta-coefficient = −2.72 × 10^−3^, FDR = 0.039).

When examining individuals stratified as having low or high baseline diversity, the high-diversity group only showed significant increases in *P. distasonis* (Appendix A; FDR < 0.05; linear mixed effects models of feature abundance by intervention time, with patient and LMP supplementation intake as random factors). In contrast, in the low-diversity group, the abundance of 15 microbial species was significantly affected by the 10% weight loss (Appendix A, FDR < 0.05). Most of them (N = 11) increased in abundance, including *A. muciniphila* (beta-coefficient = 0.036, FDR = 6.36 × 10^−4^), *M. smithii* (beta-coefficient = 0.018; FDR = 5.85 × 10^−3^) or *Intestinimonas butyriciproducens* (beta-coefficient = 0.018, FDR = 2.38 × 10^−3^) as the species with the highest effect sizes, whereas a decrease was observed for *E. rectale* (beta-coefficient = −0.049; FDR = 0.01), *S. thermophilus* (beta-coefficient = −0.028, FDR = 8.27 × 10^−3^), and bifidobacterial lineages such as *B. adolescentis* (beta-coefficient = −0.042, FDR = 0.01) and *B. bifidum* (beta-coefficient = −0.021, FDR = 0.016) (Appendix A).

In both low and high diversity groups, the species with the highest effect sizes (e.g *A. muciniphila* and *P. distasonis* for species increasing in abundance and *E. rectale* and *S. thermophilus* for species decreasing in abundance) shifted in the same direction at 10% weight loss, with significant changes observed in the group showing the highest effect sizes (p.ex. *A. mucciniphila* = FDR < 0.05 in low-diversity group only; beta-coefficient 10% weight loss vs. Baseline = 0.036 (Low-diversity group), 0.023 (high-diversity group); feature in upper-right corner of Appendix A). However, 50% of the microbial species showed specific increases at 10% weight loss in the low diversity group, including the specific increase of eight microbial species (Appendix A, FDR < 0.05 and beta-coefficient > 0 in low diversity group, FDR > 0.05 and beta-coefficient < 0 in the high diversity group), suggesting an overall high impact of dietary intervention in subjects with baseline dysbiosis.

Focusing on the eight species associated with the dysbiotic Bact2 composition at baseline (Figure 3), *P. distasonis* significantly increased after 10% weight loss in both high and low diversity groups, whereas *B. dorei* showed significant increases specifically in the high-diversity group (Appendix A). The rest of the Bact2-associated species showed no significant changes in abundance during the intervention, although a tendency was observed towards higher decreases in the group with low diversity at baseline (Appendix A).

In total, individuals with low microbiome diversity at baseline exhibited the most significant changes in bacterial species compared to individuals with high microbiome diversity at baseline.

### 3.7. LMP Supplementation, Microbial Diversity and Composition, and Bacteroides 2 Prevalence

Of the 163 individuals for which metagenomics data were available at baseline and after 10% weight loss, 116 (71%) took the LMP supplement in addition to the diet intervention, whereas 47 (29%) did not. We performed the same analysis (diversity, species, functions) in regard to LMP supplementation.

LMP consumption seemed to have significant effects in the low-diversity group. While GM diversity improvements were not statistically significant in this group (Figure 5a; *p*-value = 0.2; comparison of species richness fold-changes after 10% weight loss), the proportion of Bact2 individuals decreased among LMP consumers (38% of Bact2 individuals at baseline versus 19% of Bact2 individuals at 10% weight loss, *n* = 58, *p*-value = 6.07 × 10^−4^, Figure 5b). In contrast, the enterotype composition of non-LMP consumers was not significantly altered (Figure 5b; *n* = 31, 45% of Bact2 individuals at baseline and after 10% weight loss *p*-value = 0.54).

### 3.8. LMP Supplementation, Gut Microbial Species and Functional Module in Matched Subjects

To overcome the strongly unbalanced sizes of LMP consumers (*n* = 116) or LMP non-consumers (*n* = 47), univariate analyses of the abundances of microbial species and functions between baseline and 10% weight loss and stratified by LMP intake were carried out on matched sub-groups of individuals by age, gender and weight (*n* = 47 individuals per group) by the method of propensity scores [33].

The most striking results were a higher decrease in species from the Bacteroides lineages such as *B. vulgatus* (beta-coefficient 10% weight loss vs. baseline = −0.035 vs. −0.012; *p*-value = 0.024 vs. 0.31, in LMP consumers vs. non-consumers) or *B. fragilis* (beta-coefficient 10% weight loss vs. baseline = −0.015 vs. 1.89 × 10^−2^; *p*-value = 8.6 × 10^−3^ vs. 0.78, LMP consumers versus non-consumers) in LMP consumers after 10% weight loss. These decreases were accompanied by a higher increase in *F. prausnitzii* (Appendix A; beta-coefficient 10% weight loss vs. baseline = 0.0405 vs. 0.012; *p*-value = 8.82 × 10^−3^ vs. 0.51, LMP consumers vs. non-consumers; linear mixed-effects models of feature abundance by intervention time with patient as a random factor).

By contrast, *E. rectale* was the species that decreased more in LMP non-consumers compared to LMP consumers (Appendix A; beta-coefficient 10% weight loss vs. baseline = −0.024 vs. −0.045; *p*-value = 0.157 vs. 2.21 × 10^−3^, consumers vs. non-consumers). Finally, the abundance of *A. muciniphila* increased significantly in both LMP and non-LMP consumers (Appendix A).

On the functional level, LMP supplementation was also positively associated with modules involved in the degradation of carbohydrates such as fructose and rhamnose, and negatively associated with amino-acid degradation (Appendix A). Such associations were not found in the LMP non-consumers (Appendix A; beta-coefficients 10% weight loss vs. baseline < 0 in LMP consumers’ group; >0 in the non-consumers group; linear mixed-effects models of feature abundance by intervention time with patient as a random factor). However, butyrate production potential was diminished both in LMP consumers and LMP non-consumers.

Overall, in our cohort, LMP usage was linked to some improvements of the bacterial composition (e.g., less Bact2-linked species, or increased *F. prausnitzii* and potential impact on amino acid processing). On the other hand, this calorie-restrictive diet with high protein and low carbohydrates is associated with overall increased *A. muciniphila* and a decrease in butyrate production potential independently of LMP consumption (Appendix A).

## 4. Discussion

In a population involved in a weight management program [22], we found that individuals with obesity and overweight were stratified into four enterotypes based on gut microbiome ONT sequencing data. In these four enterotypes, we detected the Bact2 enterotype, a characteristic composition of a dysbiotic GM. This is, to our knowledge, the first time that discrete microbiome communities have been detected using this technology. At baseline, subjects from the Bact2 enterotype group displayed a lower microbiome diversity and more severe clinical profile with increased BMI and worse metabolic health markers (triglycerides, liver transaminases) compared to the other enterotypes. Interestingly, the prevalence of Bact2 was reduced at 10% weight loss during the diet intervention, but only in the group of individuals that exhibited low microbial diversity at baseline. Overall, it seems that the Bact2 profile could be an interesting target for microbiome-centered therapeutic interventions.

Given the complexity and variability of the gut microbiome, the existence of enterotypes has been a subject of controversy even though they can be linked to some clinical features. There is indeed a large inter-individual variability between subjects classified within a given enterotype [35,36]. However, several studies have reproduced the presence of enterotypes with similar compositional properties across large independent datasets from various populations or disease groups [15].

Recently, the split of Bacteroides groups into two subgroups, Bact1 and Bact2, showed that the latter could be linked to low GM gene richness and reduced stool microbial cell count (e.g., low bacterial biomass) [9,27]. For instance, in the European MetaCardis population [14], the prevalence of Bact2 enterotype increased with increased body mass index, and individuals in the Bact2 group had higher systemic inflammation [14]. Bact2 is also associated with inflammatory diseases such as ulcerative colitis, Crohn’s disease, primary sclerosing cholangitis, depression, and multiple sclerosis subtypes [16,37,38]. Bact2 is thus viewed as a gut microbiome signature with clinical relevance, however, it was not previously characterized in the context of a weight loss or dietary intervention study setting. Here, using ONT sequencing technology that has recently been adapted to be used for microbiome sequencing [25], we found that Bact2 was successfully detected and prevalent in subjects from the French “GutInside” population (19.4% of the population). We identified Bact2-related functional modules linked to amino acid degradation, central metabolism, and degradation of carbohydrates that are in line with the increased proteolytic and saccharolytic capacity of Bacteroides-enriched microbiome [39].

The Ruminococcus-enriched enterotype (Rum) was also detected in our cohort. This enterotype is characterized by a higher level of microbial richness compared to the other enterotypes as well as the presence of *F. prausnitzii*, one of the main butyrate-producing bacteria in the human gut that is frequently depleted in diseases with a low inflammatory component [40]. Intriguingly, we did not detect the Prevotella enterotype [12,41,42,43] in our cohort; however, we instead found a Bifidobacterium-enriched microbiome composition (Bif), which was significantly more prevalent in the young women of our cohort. The Bif enterotype has previously been observed in individuals from Asian populations [33]. In the context of our current study, the lack of close relatives of the Prevotella lineages in the reference genomic database used by ONT taxonomic profiling could explain this observation. Further research with additional reference genomic databases would be needed to confirm this hypothesis.

In our study, we also confirmed that the Bact2 enterotype was associated with low gene richness and clinical characteristics including more severe obesity, increased markers of insulin resistance, and sleep apnea [32]. Inflammatory markers were not collected in this cohort thus this association from previous studies could not be examined. Moreover, the lack of fresh stool samples due to the real-life settings of this study precluded the evaluation of the microbial cell density of the gut ecosystem, which is another main characteristic of the Bact2 composition. It is useful to emphasize that these observations are associative by nature, and further research would be needed to uncover a putative mechanistic link between the prevalence of Bact2 enterotype and the severity of patients’ metabolic phenotypes. Since relationships between the gut microbiome and its host are bidirectional, it remains unclear whether the gut microbiome composition is a result or a cause of the observed metabolic features.

Dietary changes are well-known to induce compositional changes in the gut microbiome [5]. Different studies have reported beneficial effects of the Mediterranean diet, and diets rich in fibers and whole grains promote the growth of beneficial gut microbes associated with short-chain fatty acid production from carbohydrates including *Faecalibacterium prausnitzii*, *Eubacterium rectale*, or different Bifidobacterium lineages [44,45,46]. In contrast, these same lineages are depleted in diets generally lacking these bacterial substrates, including a gluten-free diet [47] and protein-rich diets with low fiber content [48,49,50]. Within these diets, there is also an impact of the source of dietary protein (animal, vegetal) on microbiome and metabolome profiles [51]. In our current cohort, we evaluated the compositional changes that are observed after 10% weight loss. The calorie-restricted, high-protein low carbohydrate diet in the current study was previously shown to improve subjects’ metabolic conditions and to limit weight regain in a population from the European DIOGENES consortium [22]. We further confirmed here that the subjects’ metabolic traits were improved, in addition to significant weight losses. Interestingly, while gut microbial diversity was not significantly affected by the intervention, significant improvements were observed in individuals with a low microbial diversity pre-intervention.

In line with the reports above on dietary substrates and gut microbiome changes, we also found a decrease in firmicutes and bifidobacterial lineages at 10% weight loss, which was parallel to functional changes in decreased fiber degradation and butyrate production capabilities. Diet-induced reduction of *E. rectale* and Bifidobacterium lineages, as well as a reduced butyrate production, has already been described in subjects with obesity following high protein and low carbohydrate diets as part of a weight-loss program [52,53]. Insufficient amounts of dietary fiber in the diet of our current study could also explain the decrease in *Bifidobacterium* lineages observed. It was indeed demonstrated that oligofructose (present in dietary fiber) selectively increases the gut microbiome bifidobacterial content [54]. Along this line, *S. thermophilus* is another well-known probiotic bacterium (main yogurt starter) that stimulates carbohydrate degradation in milk-based products. Thus, a low carbohydrate high protein diet may be detrimental to these probiotic strains that rely on carbohydrates for growth. While the LMP supplement in this study population contained dietary prebiotics, it was a low concentration of inulin (27.6 mg) and fructo-oligosaccharide (2.4 mg), which is not sufficient to sustain bacterial growth as observed in previous studies [55]. In fact, the administration of 16 g/day of a mixture of inulin and oligofructose for 3 months in subjects with obesity increased the abundance of *Bifidobacterium* and *F. prausnitzii,* while decreasing *B. intestinalis*, *B. vulgatus*, and *Propionibacterium* [56]. Methanogenic lineages, such as the archaeon *M. smithii,* seemed to thrive under the dietary conditions in our study, which paralleled increasing gut microbiome methanogenic capabilities.

Despite the decrease in beneficial gut bacterial lineages associated with butyrate production, we also observed a significant increase in the abundance of other beneficial gut commensals such as *A. muciniphila* and *P. distasonis,* both having been recently shown to alleviate obesity and metabolic dysfunctions in mice [57]. *A. muciniphila* has been further associated with metabolic health status including better glucose homeostasis, blood lipid profile, and body composition after calorie restriction [58]. In fact, *A. muciniphila* has been extensively studied in mouse and human gut microbiomes and is a next-generation probiotic candidate with both proof-of-concept and clinical studies supporting its effectiveness [59]. In addition, *P. distasonis* has been successfully shown to be beneficial in decreasing weight gain, hyperglycemia, and hepatic steatosis in *ob/ob* and high-fat-fed mice [57]. However, it is unclear though whether the changes observed in our setting can contribute to weight loss and metabolic improvements. Moreover, it cannot be excluded that those changes are simply related to the higher consumption of proteins during the weight-loss intervention. In a recent paper, Bel Lassen and al. [60] examined the associations between protein intake and gut microbiome composition in two cross-sectional populations from the Netherlands and France and found that *P. distasonis* and *F. prausnitzii* abundance was positively associated with protein consumption.

The Bact2 enterotype is tightly linked to low diversity and we have previously shown in three independent studies that changes in gut microbiome diversity after a dietary or surgically-induced weight loss are affected by baseline gut microbiome diversity (i.e., pre intervention) [6,9,61]. Specifically, we have found that low-diversity subjects exhibit the most potential for gut microbiota diversity improvements, and these results prompted us to examine the relevance of such a baseline stratification in the GutInside cohort [6,9]. We confirmed the significant increase in gut microbiome diversity in subjects starting from a low diversity (below the cohort median) at baseline. We also demonstrated a significant decrease in Bact2 frequency after weight loss in this group when individuals were using the LMP supplement. No such improvement of Bact2 prevalence was found for individuals not consuming the LMP supplement, even though they significantly improved their gut microbiome diversity. In addition, LMP consumers exhibited a higher increase in *F. prausnitzii* and a lower decrease in *E. rectale* compared to their LMP non-consumer counterparts at 10% weight loss.

In contrast to individuals with low microbial diversity at baseline, we found that subjects with a high diversity (above the cohort median) exhibited a significant decrease in their gut microbiome diversity at 10% weight loss. Surprisingly, 12 subjects from this group started in Bact1 at baseline and switched to the Bact2 enterotype after 10% weight loss. This was associated with an increase in circulating uric acid when compared to those who remained Bact1 after weight loss. We can only speculate that this association may be due to specific food consumption or deviation from the proposed program. For example, high intakes of purine-rich red meat and seafood increase uric acid concentration, which has been described as a risk factor for metabolic syndrome and its components [62] which, in turn, may be associated with worse gut microbiome health. In this context, it is worth noticing that a decrease in gut microbiome diversity in parallel with improvements in metabolic health status has been previously observed in a fiber-rich dietary intervention [63]. So even if high diversity is generally considered as a feature of a healthy gut microbiome, there are exceptions that stimulate the need to identify additional and better signatures of a dysbiosis, such as the Bact2 composition described here and in other studies.

Unlike other studies that used probiotic supplements [64], we were unable to directly detect the bacterial species of the LMP supplement in the stools of individuals. This could be explained by the low concentration of the bacterial strains used. This could also be due to the low carbohydrate content of the weight loss intervention diet, which inhibits growth for probiotic bacteria, such as *Bifidobacteria*.

## 5. Conclusions

In this real-world weight management program, a high-protein, low-carbohydrate diet led to significant improvements in gut microbiome diversity and composition, with reduced Bact2 prevalence, albeit mostly in subjects with low diversity at baseline. The interest in using prebiotic, probiotic, or synbiotic supplementation to improve dysbiosis remains to be demonstrated through randomized studies. Finally, this study paves the way for future examinations of the Bact2 dysbiosis-related signature and could drive future personalized nutrition efforts to orient the microbiome towards a more favorable community composition for metabolic health.

## Figures and Tables

**Figure 1 biomedicines-10-00016-f001:**
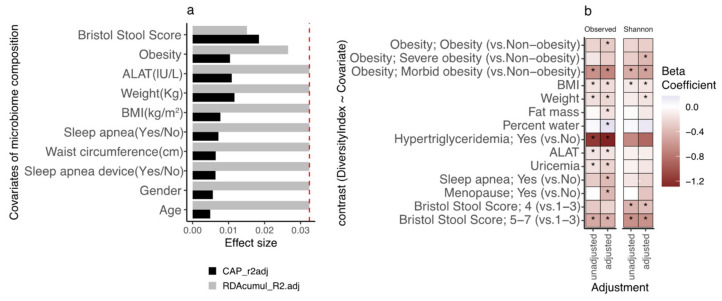
Links between clinical covariates, microbiome composition, and microbiome diversity on the GutInside baseline cohort (*n* = 263). (**a**) Variables explaining the microbiome compositional variation (distance-based redundancy analyzes, dbRDA; genus-level Bray–Curtis dissimilarity), either independently (univariate effect sizes in black; features with *p*-value < 0.05 in dbRDA) or in a multivariate model (cumulative effect sizes in grey). The cut-off for significant non-redundant contribution to the multivariate model is represented by the red line (*p*-value < 0.05 in stepwise model building). (**b**) Heatmap of beta-coefficients product of linear regression of species richness and Shannon evenness (dependent variable) vs. clinical covariates (dependent variable; y-axis) under unadjusted and adjusted design by age, gender, and center of recruitment (x-axis). Only significant contrasts are included in the figure (* = *p*-value < 0.05; Obesity: 4-level variable based on BMI ranges < 30 (non-obesity), 30 ≤ BMI < 35 (obesity); 35 ≤ BMI < 40 (severe obesity); BMI ≥ 40 (Morbid obesity)).

**Figure 2 biomedicines-10-00016-f002:**
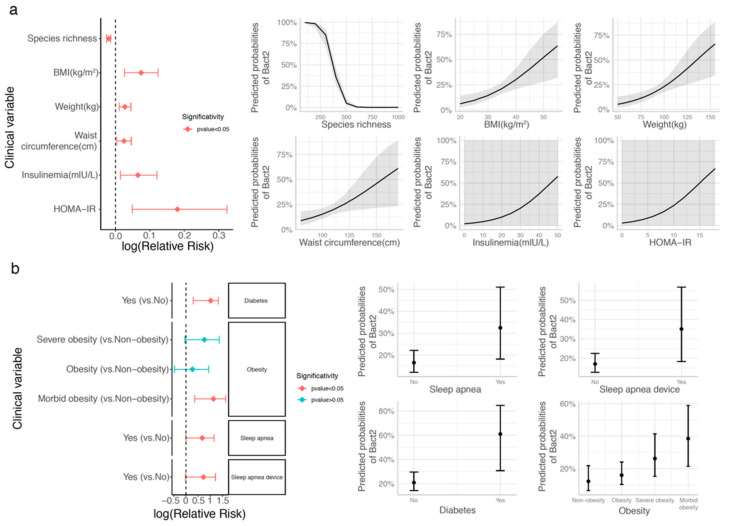
Association between Bact2 enterotype and clinical covariates in the GutInside baseline cohort (*n* = 263). Clinical covariates with significant association with Bact2 status over baseline cohort based on logistic regression analyses adjusted by age, gender, and center of recruitment. Results are represented as relative risk score intervals of Bact2 status (left side plots) for numerical (**a**) and categorical variables (**b**) with significant associations (*p*-value < 0.05). Panels on the right represent the predicted probability of Bact2 status based on logistic regression models of each clinical covariate in relative risk score panels (*p*-value < 0.05; adjusted by age, gender, and center of recruitment; Obesity = 30 ≤ BMI < 35; Severe obesity = 35 ≤ BMI < 40; Morbid Obesity = BMI ≥ 40; non-obesity = BMI < 30).

**Figure 3 biomedicines-10-00016-f003:**
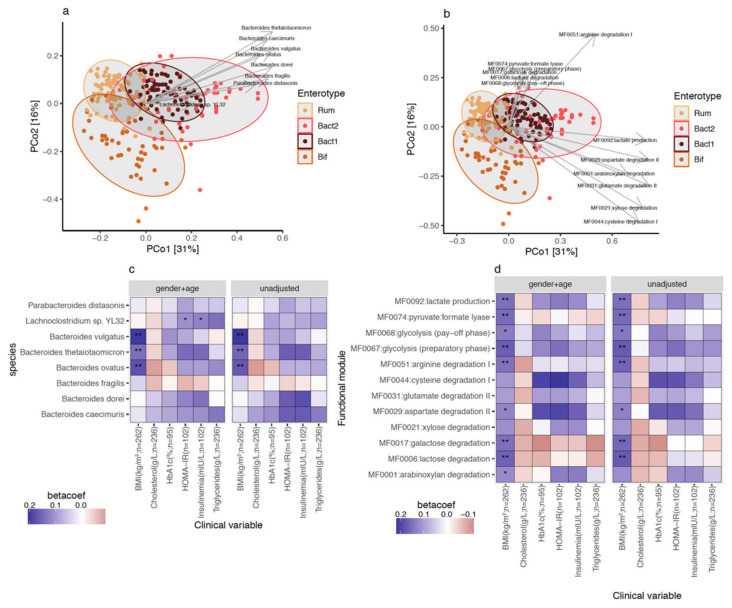
Taxonomic and functional features enriched in Bact2 microbiome composition and association with clinical variables on GutInside baseline cohort (*n* = 263). Principal coordinate analyses (PCoA) of 263 individuals of GutInside cohort colored by microbiome enterotypes with abundance vectors of bacterial species (**a**) and functional modules (**b**) enriched in Bact2 microbiome composition (FDR < 0.05, Kruskal–Wallis test of microbiome enterotypes vs. feature abundance; Cliff’s delta effect size >0 in pairwise comparisons of feature abundance in Bact2 samples vs. samples from other 3 enterotypes) fitted on the ordination plot. (**c**) Heatmap of standardized beta-coefficients showing the associations between the abundance of the 8 bacterial species enriched in Bact2 microbiome composition and clinical variables in 263 individuals of the GutInside baseline cohort based on linear regression models adjusted by different cofounding variables (* = *p*-value < 0.05; ** = FRD < 0.05; linear regression models of each bacterial specie (y-axis) vs. clinical variables (x-axis) adjusted by cofounding variables in the top of each facet). (**d**) Same as (**c**) with the 12 functional modules enriched in Bact2 microbiome composition.

**Figure 4 biomedicines-10-00016-f004:**
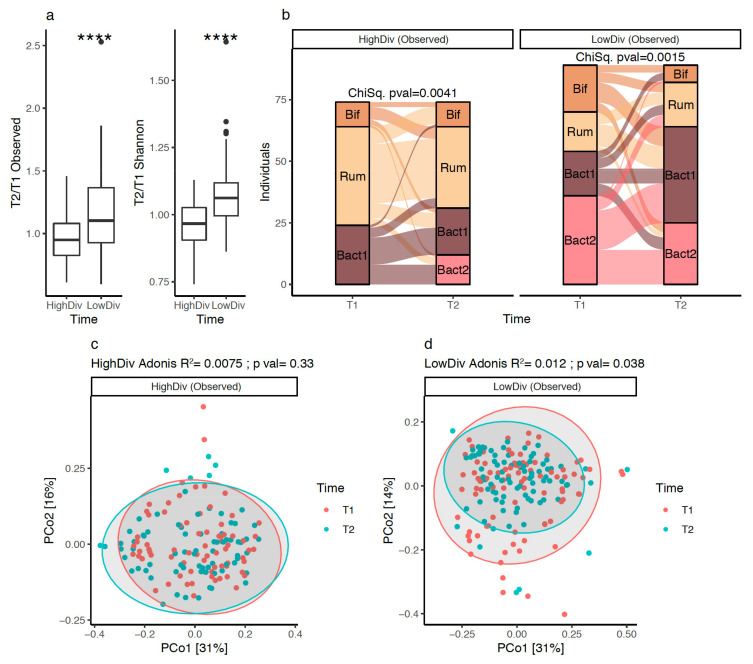
Evolution of microbiome diversity and composition at 10% weight loss in 163 individuals of GutInside study (follow-up cohort): (**a**) Boxplots of species richness and Shannon evenness fold changes distributions (10% weight loss vs. baseline; y-axis) between high and low diversity groups defined from the median of the entire baseline population (*n* = 263 individuals; 490 species). **** = *p*-value < 0.001; Wilcoxon signed-rank test). (**b**) Alluvial plots showing the evolution of enterotype composition of 163 individuals of the follow-up cohort (y-axis) between baseline (T1) and 10% weight loss (T2; x-axis) stratified by the baseline microbiome diversity of individuals (high vs. low diversity based on species richness index). *p*-value of Chi-Square test individuals within each diversity group showed in the top of each plot. (**c**) PCoA (genus-level Bray–Curtis beta-diversity) of inter-sample composition of 74 individuals in the High-diversity group between baseline (T1) and 10% weight loss (T2). Results of the PERMANOVA test to evaluate the impact of dietary intervention on microbiome composition shown on the top of the plot. (**d**) PCoA (genus-level Bray–Curtis beta-diversity) of inter-sample composition of 89 individuals in the Low-diversity group between baseline (T1) and 10% weight loss (T2). Results of the PERMANOVA test to evaluate the impact of dietary intervention on microbiome composition shown on the top of the plot.

**Figure 5 biomedicines-10-00016-f005:**
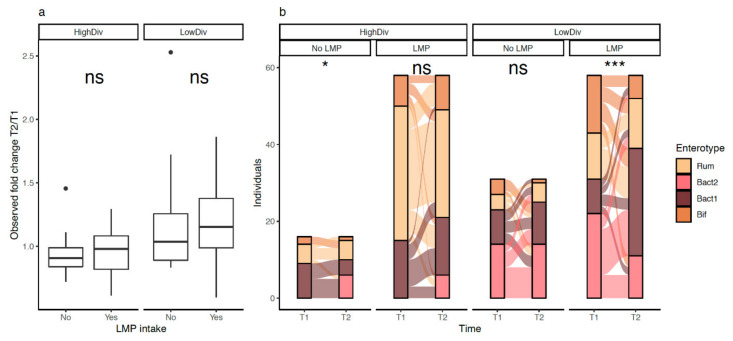
Effect of LMP supplement intake on microbiome diversity and enterotype composition across individuals of the GutInside follow-up cohort (*n* = 163). (**a**) Boxplots of species richness fold changes distributions (10% weight loss vs. baseline; y-axis) between individuals taking or not the LMP supplement stratified by the baseline microbiome diversity of individuals (high vs. low diversity) (ns = non-significant). (**b**) Alluvial plots showing the evolution of enterotype composition of 163 individuals of the follow-up cohort (y-axis) between baseline (V1) and 10% weight loss (V2; x-axis) stratified by the baseline microbiome diversity of individuals (high vs. low diversity) and the intake of LMP supplement. *p*-value of chi-square test within each diversity group showed in the top of each plot (ns = non-significant; * = *p*-value < 0.05; *** = *p*-value < 0.001). *n* = 16 (HighDiv-No LMP intake), *n* = 31 (LowDiv No LMP intake), *n* = 58 (HighDiv-LMP intake), *n* = 58 (LowDiv-LMP intake).

**Table 1 biomedicines-10-00016-t001:** Clinical parameters of the studied cohort of 263 subjects at baseline.

Clinical Parameters	All	No LMP	LMP	*p*-Value
Sample size, *n* (%)	263 (100.0)	75 (28.5)	188 (71.5)	-
Age (years)	50.6 (9.9)	50.8 (10.1)	50.5 (9.8)	ns
Sex (male), *n* (%)	55 (20.9)	21 (28.0)	34 (18.1)	ns
Height (cm)	165.8 (7.9)	167.2 (8.4)	165.3 (7.7)	ns
**Adiposity markers**				
Weight (kg)	90.5 (16.3)	93.1 (18.4)	89.4 (15.3)	ns
BMI (kg/m^2^)	32.8 (4.9)	33.2 (5.4)	32.7 (4.8)	ns
Waist circumference (cm)	106.5 (12.4)	107.9 (13.4)	106.0 (12.0)	ns
Fat mass (%)	40.7 (4.7)	40.3 (4.7)	40.9 (4.8)	ns
Fat-free mass (%)	32.0 (7.5)	30.8 (5.6)	32.0 (8.1)	ns
Calculated Basal metabolism (Kcal/day)	2217 (317)	2281 (363)	2192 (294)	ns
**Plasma Glucose homeostasis**				
Glycemia (g/L)	1.02 (0.22)	1.06 (0.23)	1.01 (0.21)	*
Insulinemia (mIU/L)	14.9 (8.7)	16.6 (10.2)	14.3 (8.0)	ns
HOMA IR	4.1 (3.1)	4.9 (4.1)	3.5 (2.7)	ns
HbA1c (%)	5.8 (0.9)	6.0 (0.9)	5.8 (0.9)	ns
**Plasma lipid homeostasis**				
Total cholesterol (nmol/L)	5.5 (1.1)	5.6 (1.0)	5.5 (1.1)	ns
Total triglycerides (nmol/L)	1.6 (0.9)	1.7 (1.2)	1.5 (0.8)	ns
HDL cholesterol (nmol/L)	1.4 (0.4)	1.4 (0.4)	1.4 (0.4)	ns
LDL cholesterol (nmol/L)	3.4 (1.0)	3.5 (0.8)	3.4 (1.1)	ns
**Liver Enzymes**				
ASAT (IU/L)	24.0 (10.0)	24.0 (9.7)	24.0 (10.2)	ns
ALAT (IU/L)	30.2 (18.3)	33.1 (20.7)	29.1 (17.1)	ns
GGT (IU/L)	41.0 (40.9)	54.3 (50.1)	36.0 (36.0)	*
**Other variables**				
Creatinine (mg/L)	8.1 (1.5)	8.1 (1.6)	8.1 (1.5)	ns
Uricemia (mg/L)	52.1 (16.3)	55.1 (11.3)	51.0 (17.6)	ns
SAS (Yes), *n* (%)	37 (14.1)	10 (13.3)	27 (14.4)	ns

Results are expressed as the mean (SD) for continuous data and *n* (%) for categorical data. *p* values result from Student’s *t* test for continuous data and Chi2 or Fisher’s exact test for categorical data between the two groups. BMI: Body Mass Index; HOMA-IR: Homeostatic Model Assessment of Insulin Resistance. ASAT: aspartate transaminase; ALAT: alanine transaminase; GGT: gamma glutamyl-transpeptidase, HDL cholesterol: high density lipoprotein, LDL cholesterol: low density lipoprotein, SAS: Sleep Apnea Syndrome; Basal metabolic rate was calculated using Black formula; ns = *p*-value > 0.05; * = *p*-value ≤ 0.05.

**Table 2 biomedicines-10-00016-t002:** Clinical parameters of the studied cohort of 163 subjects before and after weight loss.

Clinical Parameters	Before Weight Loss	After Weight Loss (-10%)	*p*-Value
Sample size, *n* (%)	163 (100.0)	-	-
Age (years)	50.8 (10.1)	-	-
Sex (male), *n* (%)	31 (19.0)	-	-
Height (cm)	165.4 (7.8)	-	-
SAS (Yes), *n* (%)	25 (15.3)	-	-
**Adiposity markers**			
Weight (kg)	90.4 (16.1)	80.4 (14.4)	****
BMI (kg/m^2^)	33.0 (4.9)	29.3 (4.4)	****
Waist circumference (cm)	106.9 (12.5)	96.3 (11.7)	****
Fat mass (%)	40.8 (4.4)	37.7 (5.0)	****
Fat-free mass (%)	32.5 (8.3)	33.2 (7.6)	****
Calculated Basal metabolism (Kcal/day)	2195 (306)	2080 (295)	****
**Plasma Glucose homeostasis**			
Glycemia (g/L)	1.03 (0.20)	0.97 (0.12)	***
Insulinemia (mIU/L)	14.0 (7.6)	10.2 (6.3)	**
HOMA IR	4.3 (3.7)	2.5 (1.9)	*
HbA1c (%)	6.2 (1.1)	5.6 (0.7)	**
**Plasma lipid homeostasis**			
Total cholesterol (nmol/L)	5.7 (1.0)	5.1 (1.0)	****
Total triglycerides (nmol/L)	1.6 (0.8)	1.1 (0.5)	****
HDL cholesterol (nmol/L)	1.4 (0.4)	1.4 (0.4)	ns
LDL cholesterol (nmol/L)	3.5 (0.9)	3.2 (0.8)	****
**Liver Enzymes**			
ASAT (IU/L)	23.8 (8.5)	21.9 (7.6)	**
ALAT (IU/L)	30.0 (18.0)	24.6 (12.1)	**
GGT (IU/L)	50.3 (53.4)	34.3 (42.2)	**
**Other variables**			
Creatinine (mg/L)	8.4 (1.5)	8.3 (1.4)	ns
Uricemia (mg/L)	53.0 (16.0)	49.8 (15.1)	**

Results are expressed as mean (SD) for continuous data and *n* (%) for categorical data. *p* values result from Student’s *t*-test paired. BMI: Body Mass Index; HOMA-IR: Homeostatic Model Assessment of Insulin Resistance. ASAT: aspartate transaminase; ALAT: alanine transaminase; GGT: gamma glutamyl-transpeptidase, HDL cholesterol: high density lipoprotein, LDL cholesterol: low density lipoprotein, SAS: Sleep Apnea Syndrome; basal metabolic rate was calculated using Black formula; ns = *p*-value > 0.05; * = *p*-value ≤ 0.05; ** = *p*-value ≤ 0.01; *** = *p*-value ≤ 0.001; **** = *p*-value ≤ 0.0001.

## Data Availability

The data presented in this study are available on request from the corresponding authors.

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
