# Peer review of "Characterization of the Gut Microbiota in Individuals with Overweight or Obesity during a Real-World Weight Loss Dietary Program: A Focus on the Bacteroides 2 Enterotype"

_biomedicines, 2021, doi:10.3390/biomedicines10010016_

Round 1

Reviewer 1 Report

The work of Rohia Alili et al. is innovative and interesting. Authors demonstrate how a bodyweight decrease may have a beneficial impact on gut microbial composition. 

I have only minor remarks:

Introduction:

  • “an important marker of ecosystem richness” (remove, redundant)
  • No mention of Phyla perturbation?
  • “host metabolic outcomes during a diet(ary) intervention”

Subject and method

  • (RNPC) To better understand the acronym insert also the name of the program in the original language
  • what kind of products are used? If there is a commercial product, please indicate brand name and nutritional facts
  • There is a little bit of confusion in the statistical analysis. First, the authors say that they use Wilcoxon rank-sum test to test median differences of continuous variables between two different groups. Then, they say that they used the Mann-Whitney for the same variables. In table 1, they say that they used Student’s t-test for continuous data. Please, decide and say clearly what kind of test they really use. 

Author Response

We thank the reviewer for finding our work of interest. Please find below the point to point answers to comments

Reviewer 2 Report

Alili et al examined changes in the gut microbiota composition and putative functions in an obese cohort who underwent a high-protein low-carbohydrates weight loss dietary regimen, with some individuals also consuming a live microorganisms supplement. The large sample size was a major strength in this study. However possibly due to the nature of a “real-world” program, there was no control group in this study and compliance data were also not available which made it difficult to interpret the relationship between gut microbiota, dietary intervention and clinical outcomes. By categorizing microbiome samples based on enterotypes at the genus level, the authors reported that the Bact2 enterotype was associated with obesity and its co-morbidities.

Major comments:

  1. It is now clear that bacteria in the same genus (or even in the same species) could vary considerably in their genomes so enterotype classification based on genus could be somewhat too broad. While genus might be the finest classification that 16S rRNA amplicon sequencing could get to, there were reports of the MinION platform being capable of analyzing microbiome composition at the species/strain resolution. Was it possible to define the enterotypes in the current study based on species or strains? How would the data look if you do not consider enterotypes at all?
  2. How did the Bact2 enterotype identified in this study compare to that in other publications?
  3. After the four enterotypes were identified, it was not clear to me how the authors decided to focus on the Bact2 enterotype for subsequent analysis. It would be helpful if the authors also include detailed analysis of all four enterotypes.
  4. What was the rationale for using observed species rather than Shannon index as the diversity index to stratify baseline microbiome samples, given that many species could be very low in abundance and might have limited physiological relevance?
  5. Were samples from LMP consumers and non-LMP consumers analyzed separately in all analyses? If not what was the rationale for pooling them?
  6. The authors may wish to expand their discussion on comparing the current results with publications on the effect of weight loss on gut microbiota. In general weight loss confers a positive effect on the gut microbiota. However as the authors noted, the dietary regimen used in the current study did not seem to support some known beneficial gut bacteria with complex carbohydrates as their preferred energy substrates. Did the current data help tease out the effect of diet and weight loss on the gut microbiota?

Minor comments:

  1. It took the research participants 27-180 days to achieve 10% weight loss. Did these participants differ in their gut microbiome?
  2. The results section could benefit from some re-organization to improve clarity.
  3. Language edits are strongly recommended.

Author Response

(The authors gave the same response as above.)

Round 2

Reviewer 2 Report

Response to the comments, additional data analyses and revision of the manuscript are noted and greatly appreciated. The added analyses significantly improved the quality of work and should be included in the final manuscript. I have no further major comments except I strongly encourage the authors to comment in the Discussion on their use of diversity as a key marker for a "dysbiotic" gut microbiota. While high diversity is generally considered as a feature of a "healthy" gut microbiota, there have been reports of exceptions. A notable example would be a gut microbiota dominated by beneficial microbes could have a low diversity. Also changes in diversity give no indication of the gut microbiota composition and therefore is not necessarily a very useful marker to assess treatment effects.

Author Response

Dear reviewer,
Please find attached, our responses to your comments and suggestions.
Best regards,
Rohia ALILI
